# Glycated haemoglobin A$_{1c}$ (HbA$_{1c}$) for detection of diabetes mellitus and impaired fasting glucose in Malawi: a diagnostic accuracy study

Sujit D Rathod,[1] Amelia C Crampin,[2] Crispin Musicha,[3] Ndoliwe Kayuni,[3] Louis Banda,[3] Jacqueline Saul,[2] Estelle McLean,[2,3] Keith Branson,[2] Shabbar Jaffar,[4] Moffat J Nyirenda[2]

[1]Department of Population Health at London School of Hygiene and Tropical Medicine, London School of Hygiene and Tropical Medicine, London, UK
[2]Department of Infectious Disease Epidemiology at London School of Hygiene and Tropical Medicine, London School of Hygiene and Tropical Medicine, London, UK
[3]Malawi Epidemiology and Intervention Research Unit, Chilumba, Malawi
[4]Department of International Public Health at Liverpool School of Tropical Medicine, Liverpool School of Tropical Medicine, Liverpool, UK

**Correspondence to**
Dr Sujit D Rathod;
sujit.rathod@lshtm.ac.uk

## ABSTRACT

**Objectives** To examine the accuracy of glycated haemoglobin A$_{1c}$ (HbA$_{1c}$) in detecting type 2 diabetes and impaired fasting glucose among adults living in Malawi.

**Design** A diagnostic validation study of HbA$_{1c}$. Fasting plasma glucose (FPG) ≥7.0 mmol/L was the reference standard for type 2 diabetes, and FPG between 6.1 and 6.9 mmol/L as impaired fasting glucose.

**Participants** 3645 adults (of whom 63% were women) recruited from two demographic surveillance study sites in urban and rural Malawi. This analysis excluded those who had a previous diagnosis of diabetes or had history of taking diabetes medication.

**Results** HbA$_{1c}$ demonstrated excellent validity to detect FPG-defined diabetes, with an area under the receiver operating characteristic (AUROC) curve of 0.92 (95% CI 0.90 to 0.94). At HbA$_{1c}$ ≥6.5% (140 mg/dL), sensitivity was 78.7% and specificity was 94.0%. Subgroup AUROCs ranged from 0.86 for participants with anaemia to 0.94 for participants in urban Malawi. There were clinical and metabolic differences between participants with true diabetes versus false positives when HbA$_{1c}$ was ≥6.5% (140 mg/dL).

**Conclusions** The findings from this study provide justification to use HbA$_{1c}$ to detect type 2 diabetes. As HbA$_{1c}$ testing is substantially less burdensome to patients than either FPG testing or oral glucose tolerance testing, it represents a useful option for expanding access to diabetes care in sub-Saharan Africa.

## INTRODUCTION

Over 400 million people live with type 2 diabetes worldwide, with more than 75% of these in low/middle-income countries.[1] While the prevalence of diabetes in sub-Saharan African is lower than in other global regions, around 6%–9%,[2 3] demographic models project that this region will experience the fastest growth rate in cases over the next 20 years.[1] Approximately 50%–66% of people with diabetes are undiagnosed,[1 4] and complications are common even after diagnosis.[5] As barriers to diabetes care become surmounted in this region,[4] focus must turn to improving the accessibility of diagnostics.

The diagnosis of diabetes has traditionally been based on the detection of elevated plasma glucose levels, either after fasting or 2 hours after an oral glucose tolerance test, or, in symptomatic individuals, after a random blood glucose check.[6] While these tests are available in sub-Saharan Africa, access to services is limited, travel times and clinic waiting times are high and integrity of the samples and quality of the measuring tools are in most cases uncertain. More reliable approaches to circumvent some of these challenges are required. Recently, the American Diabetes Association and the WHO have recommended using glycated haemoglobin A$_{1c}$ (HbA$_{1c}$) in blood to diagnose diabetes mellitus.[7 8] HbA$_{1c}$ testing provides significant practical advantages over glucose testing as it does not fluctuate appreciably and thus can be performed at any time of the day. Further, HbA$_{1c}$ testing does not require any special pretest preparations, such as overnight fasting or glucose loading.

There are acknowledged limitations when attempting to measure HbA$_{1c}$ levels which may be relevant for diagnostics in

sub-Saharan Africa.[6 8 9] It is unclear whether one can use $HbA_{1c}$ to diagnose diabetes for people who have malaria,[6] haemolytic anaemia,[10 11] sickle cell anaemia,[12] HIV infection[13 14] or who are of African descent.[15 16] Given uncertainty around the validity of $HbA_{1c}$ to diagnose type 2 diabetes and its precursor, impaired fasting glucose, in sub-Saharan Africa, we conducted a diagnostic accuracy study in Malawi, and have reported results here.

### Strengths and limitations of this study

► Only two studies have assessed the validity of glycated haemoglobin A$_{1c}$ to detect type 2 diabetes in African-origin populations; this is the first such study from Malawi.
► We used fasting plasma glucose as the reference standard, which is commonly used in clinical practice and for validation studies.
► The results of this study may not be generalisable to other populations in sub-Saharan Africa, or those with anaemia.

## DESIGN AND METHODS
### Study setting
This report uses data collected as part of a Malawi Epidemiology and Intervention Research Unit survey, which aimed to understand the burden and risk factors of non-communicable diseases in Malawi,[17] where the national prevalence of diabetes is estimated to be 5.6%.[18] Community-based cross-sectional surveys were conducted in Karonga District (May 2013–April 2016)[19] and Lilongwe City (June 2013–April 2017).[17] Karonga is a rural, low-altitude, malaria-endemic district in northern Malawi, and Lilongwe is an urban, high-altitude city in central Malawi with lower malaria prevalence.

### Recruitment and data collection
Detailed study procedures have been previously reported.[17] All adults aged 18 years and above who were usually residents in either of the study sites were eligible to participate in the parent study. All households were approached consecutively, and all residents aged ≥18 years were recruited. Recruits provided written informed consent for each separate study component (ie, standardised interview, physical measurements and blood specimen collection) and could opt out of any component. Venipuncture was conducted after a minimum 8-hour fasting period and whole blood samples were collected in sodium fluoride tubes. Tubes were stored on ice in an insulated cool box and delivered to the laboratory for processing (mean delivery time of 2.6 hours after blood collection), and glucose analysis was completed within 1 hour after processing. Quantitative determination of fasting plasma glucose (FPG) (hexokinase method) and $HbA_{1c}$ was performed using the Beckman Coulter Chemistry Analyzer AU480 system according to the manufacturer's guidelines.

FPG testing was conducted on all participant samples. $HbA_{1c}$ testing was conducted on all participants samples for which FPG results were greater than 5.6 mmol/L and in a 10% random sample of those with FPG lower than 5.6 mmol/L. The laboratory technician who conducted the $HbA_{1c}$ test was blind to the participants' clinical characteristics and FPG result. Nearly all (87%) of $HbA_{1c}$ tests were completed within 24 hours of FPG test. Cut-points recommended by WHO were used to define FPG as normal (<6.1 mmol/L), impaired fasting glucose (6.1 to 6.9 mmol/L) or diabetes (≥7.0 mmol/L).[20] Categories of body mass index (BMI) for underweight/normal (BMI <25 kg/m$^2$), overweight (25≤BMI<30) and obese (BMI ≥30) and categories for anaemia (haemoglobin <12.0 g/dL for women and <13.5 for men) were used.

### Statistical analyses
Participants who reported a past diagnosis of diabetes or history of taking diabetes medication were excluded from this analysis. First, demographic and clinical variables (ie, age, sex, location, BMI, blood pressure, plasma glucose and lipid profile and HIV serostatus) were described separately for the Karonga and Lilongwe samples. Medians and IQRs were reported for the continuous measures, as most of these measures were skewed, and proportions for categorical measures. Second, distribution of $HbA_{1c}$ level was compared by FPG result (<7.0 vs ≥7.0 mmol/L) with logistic regression. Third, the validity of $HbA_{1c}$ for diagnosing diabetes, using FPG (>7.0 mmol/L) as reference standard, was assessed by using Somers' D statistic with Harrell's C transformation to estimate the area under the receiver operating characteristics (AUROCs) curve and 95% CI. Validity statistics (ie, sensitivity, specificity and positive and negative likelihood ratios) were reported for standardised $HbA_{1c}$ scores which correspond to 1.0, 2.0, 3.0 and 4.0 SD above the sample's mean $HbA_{1c}$ score, at an 'optimal' value identified by Youden's J statistic (where the sum of the sensitivity and specificity reaches its maximum),[21] and at 6.5% (140 mg/dL), which is the value recommended by several diabetes associations. The standardised values were generated using the distribution of $HbA_{1c}$ scores from the diabetes-negative participants with $HbA_{1c}$ test results (ie, all participants with FPG ≥5.6 mmol/L and a 10% subsample of participants with plasma glucose <5.6) such that the subsampled participants were upweighted 10-fold before generating the distribution. Fourth, to consider the validity of $HbA_{1c}$ to detect diabetes across subgroups, the ROC analysis was stratified by site, sex, BMI and haemoglobin level. Within each stratum, the sensitivity, specificity and positive and negative likelihood ratio when using $HbA_{1c}$ ≥6.5% (140 mg/dL) as a cut-off score were reported. Fifth, the AUROC and 95% CI for detecting impaired fasting glucose (FPG ≥6.1 mmol/L) with $HbA_{1c}$ were estimated among those who did not have diabetes (FPG <7.0 mmol/L). Finally, the clinical characteristics were described for the subsets of participants who had FPG-defined diabetes, and for true/false-positive participants with $HbA_{1c}$ ≥6.5% (140 mg/dL). Again, medians and IQRs were reported for continuous measures and proportions for categorical measures. The analysis was conducted using Stata SE V.14.2 (StataCorp, College Station, Texas, USA) (See Stata code in online supplementary material) with complete case analysis. Aside count figures, the statistical results were weighted by the inverse probability of receiving the $HbA_{1c}$ test.

### Ethical approval
The study protocol was reviewed and approved. Participants identified as having diabetes were referred to

chronic care clinics established in conjunction with the Malawi Ministry of Health, where they initiated management per national treatment guidelines.

## Patient and public involvement

The Karonga and Lilongwe population platforms' research priorities have reflected the Malawi National Health Research Agenda, and operate through a long-standing research partnership with the Malawi National Ministry of Health and Malawi College of Medicine. The design and aims of these platforms were developed in a stakeholders' meeting in Lilongwe, the capital city of Malawi, attended by policy makers, policy implementers and researchers. The designs of the platforms were influenced by staff working closely with the communities and engaging through community meetings. Prior to launching new substudies, the research team conducts community sensitisation events with dancing, dramas

and question and answer sessions. In Karonga, village elders and other responsible community members were responsible for enumerating households, and for reporting household vital events and household migrations. In Lilongwe, block leaders were involved in sensitisation and communication with the community prior to enumeration by research staff. Preliminary results from this study have been disseminated at a conference held at the University of Malawi College of Medicine, and presentations for lay audiences, such as for the participating communities, are being prepared.

## RESULTS

As shown in the Standards for Reporting of Diagnostic Accuracy Studies (STARD) flow diagram (figure 1), 30 574 adults consented to the interview and clinical

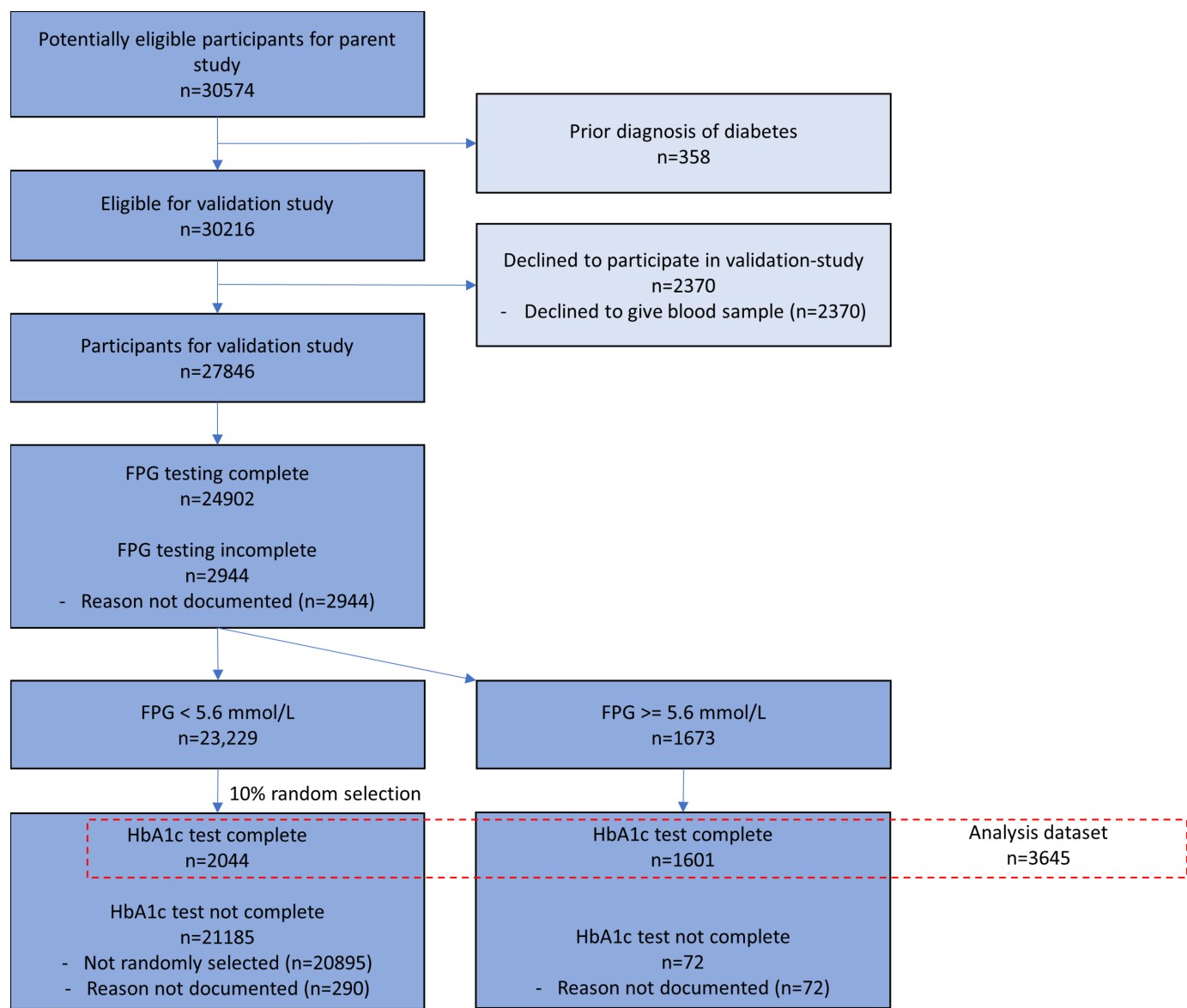

**Figure 1** Standards for Reporting of Diagnostic Accuracy Studies (STARD) diagram. FPG, fasting plasma glucose ; HbA1c, glycated haemoglobin $A_{1c}$.

**Table 1** Demographic and clinical characteristics of participants in Karonga and Lilongwe, Malawi, 2013–2017

| | Total (n=3645) Median and IQR or n and % | Karonga (n=1613) Median and IQR, or n and % | Lilongwe (n=2032) Median and IQR, or n and % |
|---|---|---|---|
| Age, years | 33 (25–44) | 34 (26–47) | 31 (24–41) |
| Female sex, % | 63.0 | 58.3 | 67.2 |
| Body mass index (BMI), kg/m$^2$ | 22.6 (20.5–26.0) | 21.8 (20.0–23.8) | 23.8 (21.2–28.0) |
| Normal BMI, % | 61.1 | 81.9 | 57.8 |
| Overweight BMI, % | 20.4 | 13.8 | 26.1 |
| Obese BMI, % | 10.5 | 4.3 | 16.0 |
| Systolic blood pressure, mm Hg | 119 (111–129.5) | 117.5 (110–127.5) | 121 (112.5–131) |
| Diastolic blood pressure, mm Hg | 72.5 (66.5–79.5) | 72 (66–78) | 73 (67–80.5) |
| Total cholesterol, mmol/L | 3.8 (3.3–4.6) | 3.8 (3.2–4.5) | 3.9 (3.3–4.6) |
| Triglycerides, mmol/L | 0.80 (0.58–1.14) | 0.81 (0.58–1.12) | 0.80 (0.59–1.18) |
| High-density lipoprotein, mmol/L | 1.12 (0.94–1.31) | 1.09 (0.91–1.28) | 1.15 (0.98–1.34) |
| Low-density lipoprotein, mmol/L | 2.53 (2.06–3.10) | 2.44 (2.00–2.96) | 2.62 (2.14–3.22) |
| Haemoglobin, g/dL | 13.7 (12.6–14.9) | 13.7 (12.6–15.0) | 13.7 (12.6–14.9) |
| Normal haemoglobin, % | 82.0 | 81.1 | 82.7 |
| Anaemia, % | 18.0 | 18.9 | 17.3 |
| HIV positive, % | 8.5 | 7.8 | 9.1 |
| HbA$_{1c}$ % | 5.3 (4.9–5.7) | 5.1 (4.7–5.5) | 5.4 (5.1–5.8) |
| HbA$_{1c}$ ≥ 6.5% (140 mg/dL), % | 7.3 | 5.6 | 8.8 |
| Fasting plasma glucose (FPG), mmol/L | 4.7 (4.3–5.0) | 4.7 (4.3–5.0) | 4.7 (4.4–5.1) |
| FPG 6.1–7.0 mmol/L, % | 1.8 | 1.7 | 1.9 |
| FPG ≥7.0 mmol/L , % | 1.7 | 1.3 | 2.1 |

Aside counts, the figures are reweighted by inverse probability of receiving HbA$_{1c}$ testing.
HbA$_{1c}$, glycated haemoglobin A$_{1c}$.

examination in the parent study, of whom 27 846 also provided consent for blood specimen collection, as was required for this substudy. No participant experienced a severe adverse event due to the specimen collection. This analysis includes the 3645 participants (1613 in Karonga and 2032 in Lilongwe) who had both FPG and HbA$_{1c}$ levels measured, and who did not report prior diagnosis of diabetes by a medical professional.

### Demographic and clinical characteristics
The demographic and clinical characteristics of analysis participants are reported in table 1.

For the 3645 participants, the median age was 33 years (IQR 24–44) and 63% were women. The median HbA$_{1c}$ level was 5.3% (IQR 4.9–5.7) and 7.3% had HbA$_{1c}$ level of 6.5% (140 mg/dL) or higher. The median FPG level was 4.7 mmol/L (IQR 4.3–5.0); 1.8% of participants had an FPG level consistent with impaired fasting glucose and another 1.7% with diabetes.

### Diagnostic validity of HbA$_{1c}$
For every percentage unit increase in HbA$_{1c}$ score, there was an almost threefold increase in the odds of having FPG-defined diabetes (OR 2.80, 95% CI 2.21 to 3.53,

R$^2$ 0.371, Wald X$^2$(1) 75.5, p<0.001). Further, HbA$_{1c}$ demonstrated excellent validity to detect for FPG-defined diabetes, with an AUROC of 0.92 (95% CI 0.90 to 0.94). Validity statistics are presented for a range of HbA$_{1c}$ cut-points in table 2.

An HbA$_{1c}$ value of 6.0% corresponded to 1.0 SD above the mean in the diabetes-negative sample's distribution of HbA$_{1c}$ scores; at this cut-point, 84.4% of people with diabetes would test positive (sensitivity) and 86.9% of people without diabetes would test negative (specificity). At the commonly recommended HbA$_{1c}$ threshold of 6.5% (140 mg/dL), sensitivity was 78.7% and specificity was 94.0%. Youden's J was at an HbA$_{1c}$ value of 6.6%, where sensitivity was 78.3% and specificity was 94.6%. Stratum-specific AUROCs (table 3) ranged from a minimum of 0.86 for participants with anaemia to a maximum of 0.94 for participants in Lilongwe. With HbA$_{1c}$ ≥6.5% (140 mg/dL), participants with anaemia had 61.5% sensitivity, 96.4% specificity and a positive likelihood ratio of 17.4, and the corresponding figures for participants in Lilongwe were 70.2%, 95.4% and 15.2.

Among participants who did not have FPG-defined diabetes, the HbA$_{1c}$ score had average validity to detect

**Table 2** Sensitivity, specificity and positive and negative likelihood ratios for detecting type 2 diabetes at selected glycated haemoglobin A$_{1c}$ (HbA$_{1c}$) thresholds for participants in Karonga and Lilongwe, Malawi, 2013–2017

| HbA$_{1c}$ threshold | | | | | | |
|---|---|---|---|---|---|---|
| % | mg/dL | SD | Sensitivity (%) | Specificity (%) | Positive likelihood ratio | Negative likelihood ratio |
| 6.0 | 126 | +1.0 | 84.4 | 86.9 | 6.5 | 0.18 |
| 6.5* | 140 | +1.7 | 78.7 | 94.0 | 13.3 | 0.23 |
| 6.6† | 143 | +1.8 | 78.3 | 94.6 | 14.5 | 0.23 |
| 6.7 | 146 | +2.0 | 76.5 | 96.0 | 19.0 | 0.25 |
| 7.4 | 166 | +3.0 | 66.3 | 98.5 | 43.3 | 0.34 |
| 8.1 | 186 | +4.0 | 54.8 | 99.5 | 117.4 | 0.45 |

Figures are reweighted by the inverse probability of receiving HbA$_{1c}$ testing.
*Diagnostic threshold recommended by American Diabetes Association, European Association for the Study of Diabetes, International Diabetes Federation and WHO.
†Threshold identified by Youden's J statistic
SD, standard deviations from the mean HbA$_{1c}$ score among diabetes-negative participants.

impaired fasting glucose (AUROC 0.70, 95% CI 0.67 to 0.73).

### Clinical characteristics of participants meeting diagnostic criteria for diabetes

The clinical characteristics of participants who had FPG-defined diabetes and for participants who had true/false-positive results are presented in table 4.

The participants who had FPG-defined diabetes (FPG ≥7.0 mmol/L) had a median age of 50 years (IQR 41–60) and 65.5% were women. The median BMI was 27.9 kg/m$^2$ (IQR 24.2–32.5) and the median systolic blood pressure 133.5 mm Hg (IQR 121.5–153). The characteristics of true-positive participants (HbA$_{1c}$ ≥6.5% (140 mg/dL) and FPG ≥7.0 mmol/L) and the

false-positive participants (HbA$_{1c}$ ≥6.5% (140 mg/dL) and FPG <7.0 mmol/L) are also described in table 4.

### DISCUSSION

In a large, multisite sample of adults in Malawi, we found that HbA$_{1c}$ is highly predictive of FPG-defined type 2 diabetes, a relationship which was consistent across several subgroups. HbA$_{1c}$ was less predictive for impaired fasting glucose. Using a cut-off value of HbA$_{1c}$ 6.5% (140 mg/dL) to detect diabetes, there were demographic and clinical differences between true-positive and false-positive participants.

This analysis is one of the first to compare HbA$_{1c}$ as a stand-alone test to detect type 2 diabetes in a black

**Table 3** Overall and stratum-specific area under the receiver operating characteristic (AUROC) curves for detection of type 2 diabetes with glycated haemoglobin A$_{1c}$ (HbA$_{1c}$) among adults in Malawi, 2013–2017

| | AUROC (95% CI) | Sensitivity (%)* | Specificity (%)* | Positive likelihood ratio* | Negative likelihood ratio* |
|---|---|---|---|---|---|
| Overall | 0.92 (0.90 to 0.94) | 78.7 | 94.0 | 13.3 | 0.23 |
| Site | | | | | |
| Karonga | 0.88 (0.94 to 0.92) | 70.2 | 95.4 | 15.2 | 0.31 |
| Lilongwe | 0.94 (0.92 to 0.96) | 83.3 | 92.9 | 11.7 | 0.18 |
| Sex | | | | | |
| Female | 0.93 (0.91 to 0.95) | 79.1 | 95.2 | 16.5 | 0.22 |
| Male | 0.89 (0.85 to 0.93) | 77.6 | 92.1 | 9.9 | 0.24 |
| Body mass index | | | | | |
| Normal | 0.87 (0.83 to 0.92) | 70.1 | 95.1 | 14.5 | 0.31 |
| Overweight | 0.93 (0.89 to 0.96) | 84.5 | 91.4 | 9.9 | 0.17 |
| Obese | 0.92 (0.89 to 0.95) | 80.4 | 90.6 | 8.6 | 0.22 |
| Haemoglobin | | | | | |
| Normal | 0.93 (0.91 to 0.95) | 81.4 | 93.5 | 12.6 | 0.20 |
| Anaemia | 0.86 (0.79 to 0.93) | 61.5 | 96.4 | 17.4 | 0.40 |

Figures are reweighted by the inverse probability of receiving HbA$_{1c}$ testing.
*At HbA$_{1c}$ ≥6.5%.

**Table 4** Clinical characteristics of participants identified as having diabetes by fasting blood glucose and for true/false-positive participants by glycated haemoglobin A$_{1c}$ (HbA$_{1c}$) ≥6.5% in Karonga and Lilongwe, Malawi, 2013–2017

| | Type 2 diabetes positive by FPG ≥7.0 mmol/L (IQR) | Type 2 diabetes true positive by HbA$_{1c}$ ≥6.5%* (IQR) | Type 2 diabetes false positive by HbA$_{1c}$ ≥6.5%*† (IQR) |
|---|---|---|---|
| Age, years | 50 (41–60) | 51 (43–61) | 35 (23–51) |
| Female sex, % | 65.6 | 66.3 | 51.0 |
| Body mass index, kg/m$^2$ | 27.9 (24.2–32.5) | 28.2 (24.8–32.5) | 24.8 (20.9–28.3) |
| Systolic blood pressure, mm Hg | 133.5 (121.5–153) | 135 (123–156) | 125 (115–136) |
| Diastolic blood pressure, mm Hg | 81.5 (74.5–89.5) | 81.5 (75.0–90.5) | 74.5 (68.5–82.5) |
| Total cholesterol, mmol/L | 4.9 (4.1–5.7) | 5.0 (4.3–5.8) | 4.2 (3.5–5.2) |
| Triglycerides, mmol/L | 1.7 (1.2–2.3) | 1.8 (1.2–2.5) | 0.9 (0.6–1.5) |
| High-density lipoprotein, mmol/L | 1.04 (0.89–1.22) | 1.03 (0.88–1.20) | 1.15 (0.97–1.29) |
| Low-density lipoprotein, mmol/L | 3.38 (2.69–4.09) | 3.52 (2.89–4.22) | 2.82 (2.36–3.56) |
| Haemoglobin, g/dL | 13.9 (12.9–14.9) | 13.9 (12.9–15.0) | 14.2 (13.1–15.1) |
| HbA$_{1c}$, % | 8.4 (6.9–11.3) | 9.6 (7.7–11.8) | 6.9 (6.6–7.4) |
| FPG, mmol/L | 9.5 (7.7–14.2) | 10.7 (8.3–15.4) | 4.8 (4.5–5.1) |

Aside counts, figures are weighted by the inverse probability of receiving HbA$_{1c}$ testing.
*Diagnostic threshold recommended by American Diabetes Association, European Association for the Study of Diabetes, WHO and International Diabetes Federation.
FPG, fasting plasma glucose.

African population in sub-Saharan Africa, and the first from Malawi. In Uganda, Mayega et al found that HbA$_{1c}$ had moderate validity (AUROC 0.75) for detecting FPG-defined diabetes,[22] which was lower than found by Hird et al in South Africa (AUROC 0.95). These findings from sub-Saharan Africa complement meta-analyses of studies in East Asia, Middle East and Europe which show that HbA$_{1c}$ is a valid test for detecting diabetes[23–25] across different ethnic groups.

There were several notable demographic and clinical differences between participants who were true versus false positives for FPG-defined diabetes at HbA$_{1c}$ ≥6.5% (140 mg/dL). Compared with participants who had HbA$_{1c}$ ≥6.5% (140 mg/dL) and FPG <7.0 mmol/L, those who had both FPG ≥7.0 and HbA$_{1c}$ ≥6.5% (140 mg/dL) were more likely to be women, to be older and to have higher BMI and blood pressure. These findings are consistent with data from the Uganda where the AUROC for all participants was 0.75, but was significantly higher at 0.90 for overweight participants and 0.98 for obese participants. This suggests that HbA$_{1c}$ is particularly useful in in detecting diabetes in individuals with insulin resistance or metabolic syndrome (eg, constellation of high BMI, hypertension, dyslipidaemia). In contrast, the false-positive group (HbA$_{1c}$ ≥6.5% (140 mg/dL) and FPG <7.0 mmol/L) lacked features of metabolic syndrome. The false-positive group is likely to include individuals with isolated postprandial hyperglycaemia, rather than fasting hyperglycaemia (perhaps from pancreatic deficiency rather than insulin resistance), which would require oral glucose tolerance testing (OGTT) to demonstrate. Notably, HbA$_{1c}$ has been shown to strongly predict OGTT-defined diabetes in South Africa, where

Hird et al estimated the AUROC at 0.94.[26] The clinical and demographic differences between true-positive and false-positive participants observed here, though interesting, remain descriptive and will require confirmation in larger studies.

We found that HbA$_{1c}$ had poor predictive ability for detecting impaired fasting glucose. This is consistent with other studies in Africa and evidence from a recent meta-analysis by Kodama et al.[22 25 26] Accordingly, evidence strongly suggests that HbA$_{1c}$ may not be appropriate for detecting impaired fasting glucose in this population.

Stratum-specific AUROC analysis indicated largely consistent values across strata (ie, by sex, site and BMI), which is evidence of suitability of HbA$_{1c}$ to detect diabetes across a range of groups. However, AUROC values were lowest for those with anaemia, which appear to be driven by a drop in sensitivity. Different forms of anaemia (eg, chronic or acute) affect the integrity and quantity of HbA1c-carrying red blood cells, resulting in HbA$_{1c}$ having diminished utility as a stand-alone diagnostic tool among individuals with anaemia.[11] For example, Son et al also found that the AUROC for HbA$_{1c}$ to detect OGTT-defined diabetes in South Korea was lower for adults with anaemia (0.86 vs 0.88).[10] Future research will be required to determine how to control anaemia in sub-Saharan Africa so that HbA$_{1c}$ testing maintains its validity.

We used FPG rather than OGTT as the reference standard for diabetes in this analysis. A 2013 meta-analysis of 13 cohort studies found that FPG is highly correlated to present and to future diabetes,[25] though none of the studies identified were from sub-Saharan Africa. A 2015 meta-analysis of diabetes measures found 27 validation studies of HbA$_{1c}$ using FPG-defined diabetes as the

diagnostic standard, compared with nine using OGTT.[27] FPG is typically preferred for validation studies due to its practical advantages, though it is unclear what proportion of individuals with diabetes is missed when FPG testing is not paired with OGTT. While FPG is highly correlated to present and to future diabetes,[24] the frequency of OGTT-derived isolated postprandial hyperglycaemia or its clinical course is not well defined in sub-Saharan Africa, and will require further research. It is worth noting that of the two $HbA_{1c}$ validation studies conducted among black populations in sub-Saharan Africa, Hird *et al* used both FPG and OGTT as diagnostic standards, with consistent results (AUROC 0.95 and 0.94, respectively).[26]

Not collecting OGTT data constitutes an important limitation of our study. In addition, although the overall sample size was large and the overall AUROC estimate was precise, the subgroups had fewer diabetes cases which resulted in our conducting of descriptive rather than hypothesis-confirming analysis of subgroup validity. Another limitation is that we did not assess the cause or type of anaemia, which would have had differential effects on $HbA_{1c}$ as well as relevant implications for clinical practice. A final limitation concerns the decay in glucose levels which occurs when blood is stored in fluoride tubes. The reduction in variation in glucose levels was independent of OGTT result, and so would have resulted our underestimating the AUROCs for diabetes and for impaired fasting glucose.

Demographic models indicate that sub-Sahara will experience a substantial increase in diabetes prevalence in the coming years, which will require urgent strategies to scale up detection and treatment in order to increase access to care. As $HbA_{1c}$ testing is less burdensome to patients than FPG and OGTT, it represents a useful option for expanding access to diabetes care. This will become particularly important as countries in sub-Saharan Africa allocate increasing resources to the health sector, and as operating costs for the $HbA_{1c}$ test reduce over the next few years.

**Acknowledgements** We are grateful to the Karonga and Lilongwe communities, participants and traditional authorities for their engagement in this work. This study would also not have been possible without the encouragement and support of the Malawi Ministry of Health, particularly the Karonga and Lilongwe District Health Offices.

**Contributors** MJN, SJ and ACC conceived the study. SJ, ACC, CM and NK designed the protocols and implemented the study. LB designed the laboratory protocols and laboratory quality control and implemented the laboratory work. JS and KB designed the data management protocols. EM designed the data cleaning protocols. SDR conducted data analysis. All authors wrote the manuscript.

**Funding** The work is supported primarily by the Wellcome Trust, United Kingdom (grant 098610Z/12/Z).

**Competing interests** None declared.

**Patient consent** Not required.

**Ethics approval** Malawi National Health Sciences Research Committee, and London School of Hygiene and Tropical Medicine (UK) Ethics Committee.

**Provenance and peer review** Not commissioned; externally peer reviewed.

**Data sharing statement** Study data are available on request from the Principle Investigators, Moffat Nyirenda (moffat.nyirenda@lshtm.ac.uk) and Amelia Crampin (mia.crampin@lshtm.ac.uk); the participant consent allows for data sharing with research collaborators. The statistical analysis code is available here as supplementary material.

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
