## [Reviewer comments · BMJ Open]

ARTICLE DETAILS

TITLE (PROVISIONAL)	Glycated hemoglobin A1c (HbA1c) for detection of diabetes mellitus and impaired fasting glucose in Malawi: a diagnostic accuracy study
AUTHORS	Rathod, Sujit; Crampin, Amelia; Musicha, Crispin; Kayuni, Ndoliwe; Banda, Louis; Saul, Jacqueline; McLean, Estelle; Branson, Keith; Jaffar, Shabbar; Nyirenda, Moffat

VERSION 1 – REVIEW

REVIEWER	Ruth Hughes Otago University, New Zealand
REVIEW RETURNED	15-Jan-2018

GENERAL COMMENTS	This is an important study that examines whether HbA1c measurement is a valid way of screening for type 2 diabetes in Malawi. Overall the paper is well written and interesting to read. My comments are as follows: 1. Abstract – objectives and design statement both need to be clearer. At present the ‘objective’ I read as all background and the ‘design’ as mainly objective.2. Methods – please state the HbA1c criteria used to define prediabetes in your study.3. Methods – please provide more detail on how the fasting plasma glucose was analysed, the type of tubes used, how the samples were handled (e.g. immediately separated, put on ice, the delay between venesection and analysis). This can all impact on the degree of glucose ‘decay’ prior to analysis and this needs to be discussed as a potential limitation as it might have led to a proportion of the ‘false positive’ HbA1c results.4. Results – too much of table 4 is reiterated in the text. There doesn’t appear to have been any statistical comparisons made on the data presented in table 4. Please can you include p values as evidence that the groups are different?5. Discussion – as per point 3 above and also please can you discuss other limitations of using a fasting glucose as the ‘gold standard’. For example, in the population studied how many individuals with type 2 diabetes would a FPG test miss compared with an OGTT?6. Discussion – Please discuss in more detail how anaemia might influence the HbA1c result (eg increase HbA1c? by how much?)
---

	7. Discussion - you have not addressed the limitations of your study, some of which are mentioned in point 5 above. 8. Consistent use of the greater or equals sign please – e.g. it is missing in the ‘design’ section of the abstract and instead reads >=; page 6 line 21 and 22 should it not be great or equal to 7?
--	--

REVIEWER	Katherine Ogurtsova Statistician, Modeller Institute for Health Services Research and Health Economics German Diabetes Center (DDZ) Leibniz Center for Diabetes Research at Heinrich Heine University Düsseldorf Germany
REVIEW RETURNED	17-Jan-2018

GENERAL COMMENTS	I had a chance to review manuscript “Glycated hemoglobin A1c (HbA1c) for detection of pre-diabetes and diabetes mellitus in Malawi: a diagnostic accuracy study” written by Rathnod S, et al, and found the manuscript clear in results and easy to read. The authors studied the diagnostic accuracy of the HbA1c test and its power in detecting diabetes in Malawi population under the restricted resources environment. Moreover, I think that the research in diabetes, or in NCD in general, in sub-Saharan African countries is urgently (vitaly) needed. The strong features of this study are a community-based big sample, diverse settings (urban vs. rural) with distinct specific conditions (malaria-endemic vs. low malaria prevalence), a reliable protocol. Overall the manuscript is worthy for publication, but I have some comments which I hope will improve the quality of the paper, which I have set out below.  1. The authors used “pre-diabetes” concept and its definition based on ADA standards. At the same time, the diagnostic accuracy of HbA1c in regards to “pre-diabetes” was attested on, in fact, the impaired fasting glucose (IFG) as a reference point. While the “pre-diabetes” concept may be regarded as disputable and controversial (N. Bansal. Prediabetes diagnosis and treatment: A review. World J Diabetes. 2015 Mar 15; 6(2): 296–303. Published online 2015 Mar 15. doi: 10.4239/wjd.v6.i2.296; PMID: PMC4360422), I think that it would be better not to use “pre-diabetes” term in the manuscript, and mention only “impaired fasting glucose” and its overlapping with HbA1c abnormalities. At the same time, the authors reasonably labeled diabetes diagnosis studied in the manuscript as “FPG-defined diabetes” that clarified the limitation of a study protocol and made the results much more straightforward. 2. The authors described the clinical characteristics for the subjects who had FPG-defined diabetes and for true- and false-positive participants by HbA1c criteria and claimed that they differed. Did the authors suggested during the manuscript writing to compare the groups by employing some statistical methods? 3. Would it be possible to show the absolute number of people in each subgroup in Tables 2-4? It’s unclear from the text and tables how much individuals with positive/negative results were identified. 4. Also, some details about log-regression FPG-defined ~ HbA1c level would be appreciated (R2, F value, degrees of freedom,
---

and significance level (p), betta-coefficients).

VERSION 1 – AUTHOR RESPONSE

Reviewer comment	Response	Tracked changes
Editorial Comments and Requests:		
- Abstract: “A key research question for type 2 diabetes control in sub-Saharan Africa concerns improving the accessibility of diagnostics” – Can you improve the clarity of this sentence? What research question are you referring to?	We have revised the Abstract Objectives per feedback from Reviewer 1 as follows: “To examine the accuracy of glycated hemoglobin A_{1c} (HbA_{1c}) in detecting type 2 diabetes and impaired fasting glucose among adults living in Malawi.”	
- Please improve the methods sections of the abstract, which are currently not very detailed/ informative. We would suggest using the sub-headings recommended in our Instructions for Authors for research articles. See: http://bmjopen.bmj.com/pages/authors/#research_articles	We have reviewed the Instructions for Authors, and accordingly have revised the Objectives, Design and Participants sections of the Abstract.	
- On page 5 you say: “Adults could provide informed consent..” Can you please improve the clarity of this sentence? Was written informed consent obtained from all participants?	We have clarified this statement as follows: “Recruits provided written informed consent for each separate study component (i.e. standardized interview, physical measurements and blood specimen collection) and could opt out of any component.”	
Reviewer: 1 Reviewer Name: Ruth Hughes Institution and Country: Otago University, New Zealand Competing Interests: none declared		
This is an important study that examines whether HbA1c measurement is a valid way of screening for type 2 diabetes in Malawi. Overall the paper is well written and interesting to read. My comments are as follows:		
1. Abstract – objectives and design statement both need to	We have revised the	

be clearer. At present the 'objective' I read as all background and the 'design' as mainly objective.	Objectives and Design sections of the Abstract as follows: “ Objectives: To examine the accuracy of glycated hemoglobin A1c (HbA1c) in detecting type 2 diabetes and impaired fasting glucose among adults living in Malawi. Design: A diagnostic validation study of HbA1c. Fasting plasma glucose (FPG) ≥ 7.0 mmol/L was the reference standard for Type 2 diabetes, and FPG between 6.1 and 6.9 mmol/L as impaired fasting glucose.”	
2. Methods – please state the HbA1c criteria used to define prediabetes in your study.	Per feedback from Reviewer 2, we have replaced the term “pre-diabetes” with “impaired fasting glucose” throughout. In the Abstract we have included the criteria: “Fasting plasma glucose (FPG) ≥ 7.0 mmol/L was the reference standard for Type 2 diabetes, and FPG between 6.1 and 6.9 mmol/L as impaired fasting glucose.” And also in the Methods section: “The cut-points recommended by WHO were used to define fasting plasma glucose (FPG) as normal (< 6.1 mmol/L), impaired fasting	

	glucose (6.1 to 6.9 mmol/L) or diabetes (≥ 7.0 mmol/L) (20).” We have also corrected the reference for the cut points for impaired fasting glucose, which is from WHO guidelines rather than ADA guidelines.	
3. Methods – please provide more detail on how the fasting plasma glucose was analysed, the type of tubes used, how the samples were handled (e.g. immediately separated, put on ice, the delay between venesection and analysis). This can all impact on the degree of glucose ‘decay’ prior to analysis and this needs to be discussed as a potential limitation as it might have led to a proportion of the ‘false positive’ HbA1c results.	We have added the following details to the Methods section: “Venipuncture was conducted after a minimum 8-hour fasting period and whole blood samples were collected in sodium fluoride tubes. Tubes were stored on ice in an insulated cool box and delivered to the laboratory for processing (mean delivery time of 2.6 hours after blood collection), and glucose analysis was completed within 1 hour after processing”	
4. Results – too much of table 4 is reiterated in the text. There doesn’t appear to have been any statistical comparisons made on the data presented in table 4. Please can you include p values as evidence that the groups are different?	We have cut text from this section about Table 4 results. We did not have any specific hypotheses regarding clinical characteristics of screen-positive vs screen-negative participants, and so did not conduct hypothesis tests. This section was meant to be a descriptive analysis, as stated in the Methods section: “Finally, the clinical characteristics were described for the subsets of participants	

	who had FPG-defined diabetes, and for true- and false-positive participants with $HbA_{1c} \geq 6.5\%$ (48 mmol/mol).” To reinforce this point we have added the following text to the Discussion: “The clinical and demographic differences between true- and false-positive participants observed here, though interesting, remain descriptive and will require confirmation in larger studies.”	
5. Discussion – as per point 3 above and also please can you discuss other limitations of using a fasting glucose as the ‘gold standard’. For example, in the population studied how many individuals with type 2 diabetes would a FPG test miss compared with an OGTT?	We have incorporated a section in the Discussion about using FPG vs OGTT as the diagnostic standard. While it is beyond the scope of this study to compare the differential prevalence using these diagnostic standards, we point out: “FPG is typically preferred for validation studies due to its practical advantages, though it is unclear with proportion of individuals with diabetes are missed when FPG testing is not paired with OGTT. While FPG is highly correlated to present and to future diabetes, the frequency of OGTT-derived isolated postprandial hyperglycemia or its clinical course is not well defined in sub-Saharan	

	Africa, and will require further research. It is worth noting that of the two HbA_{1c} validation studies conducted among black populations in sub-Saharan Africa, Hird et al used both FPG and OGTT as diagnostic standards, with consistent results (AUROC 0.95 and 0.94, respectively)”	
6. Discussion – Please discuss in more detail how anaemia might influence the HbA1c result (eg increase HbA1c? by how much?)	We agree that this finding warrants further discussion. We have revised the Discussion section concerning subgroup analysis, which reads as follows: “Different forms of anaemia (e.g. chronic or acute) affect the integrity and quantity of HbA_{1c}-carrying red blood cells, resulting in HbA_{1c} having diminished utility as a stand-alone diagnostic tool among individuals with anaemia. For example, Son et al also found that the AUROC for HbA_{1c} to detect OGTT-defined diabetes in South Korea was lower for adults with anaemia (0.86 vs 0.88). Future research will be required to determine how to control anaemia in sub-Saharan Africa so that HbA_{1c} testing maintains its validity.”	
7. Discussion - you have not addressed the limitations of your study, some of which are mentioned in point 5 above.	We have included a paragraph for limitations: “Not collecting OGTT data constitutes an important limitation of our study. In addition, although the overall sample size was large and the overall	

	AUROC estimate was precise, the subgroups had fewer diabetes cases which resulted in our conducting descriptive rather than hypothesis-confirming analysis of subgroup validity. Another limitation is that we did not assess the cause or type of anaemia, which would have had differential effects on HbA1c as well as relevant implications for clinical practice.”	
8. Consistent use of the greater or equals sign please – e.g. it is missing in the ‘design’ section of the abstract and instead reads >=; page 6 line 21 and 22 should it not be great or equal to 7?	We have corrected these inequalities. In Abstract/Design: “Fasting plasma glucose (FPG) ≥ 7.0 mmol/L was the reference standard for Type 2 diabetes, and FPG between 6.1 and 6.9 mmol/L as impaired fasting glucose.” In Methods: “Second, distribution of HbA_{1c} level was compared by FPG result (<7.0 vs ≥ 7.0 mmol/L) with logistic regression.”	
Reviewer: 2 Reviewer Name: Katherine Ogurtsova Institution and Country: Statistician, Modeller, Institute for Health Services Research and Health Economics, German Diabetes Center (DDZ), Leibniz Center for Diabetes Research at Heinrich Heine University Düsseldorf, Germany Competing Interests: None declared		
I had a chance to review manuscript “Glycated hemoglobin A1c (HbA1c) for detection of pre-diabetes and diabetes mellitus in Malawi: a diagnostic accuracy study” written by Rathnod S, et al, and found the manuscript clear in results and easy to read. The authors studied the diagnostic		

accuracy of the HbA1c test and its power in detecting diabetes in Malawi population under the restricted resources environment. Moreover, I think that the research in diabetes, or in NCD in general, in sub-Saharan African countries is urgently (vitaly) needed. The strong features of this study are a community-based big sample, diverse settings (urban vs. rural) with distinct specific conditions (malaria-endemic vs. low malaria prevalence), a reliable protocol. Overall the manuscript is worthy for publication, but I have some comments which I hope will improve the quality of the paper, which I have set out below.		
1. The authors used “pre-diabetes” concept and its definition based on ADA standards. At the same time, the diagnostic accuracy of HbA1c in regards to “pre-diabetes” was attested on, in fact, the impaired fasting glucose (IFG) as a reference point. While the “pre-diabetes” concept may be regarded as disputable and controversial (N. Bansal. Prediabetes diagnosis and treatment: A review. World J Diabetes. 2015 Mar 15; 6(2): 296–303. Published online 2015 Mar 15. doi: 10.4239/wjd.v6.i2.296; PMID: PMC4360422), I think that it would be better not to use “pre-diabetes” term in the manuscript, and mention only “impaired fasting glucose” and its overlapping with HbA1c abnormalities. At the same time, the authors reasonably labeled diabetes diagnosis studied in the manuscript as “FPG-defined diabetes” that clarified the limitation of a study protocol and made the results much more straightforward.	We agree and have replaced “pre-diabetes” with “impaired fasting glucose” throughout.	
2. The authors described the clinical characteristics for the subjects who had FPG-defined diabetes and for true- and false-positive participants by HbA1c criteria and claimed that they differed. Did the authors suggested during the manuscript writing to compare the groups by employing some statistical methods?	Given the paucity of true positive participants, we realized early in the manuscript writing that we had limited statistical power for sub-group analysis. As such, the section considering clinical characteristics of true- vs false-positive participants was designed as a descriptive analysis rather than for hypothesis-testing. Per feedback from Reviewer 1, we will make this choice clear in the Discussion.	

3. Would it be possible to show the absolute number of people in each subgroup in Tables 2-4? It's unclear from the text and tables how much individuals with positive/negative results were identified.	We appreciate that absolute numbers are usually helpful. However, due to the use of sub-sampling and subsequent re-weighting, reporting absolute numbers (i.e. unweighted counts) would give an incomplete or even misleading picture of our findings. In Tables 2-4 some participants are weighted as 1 observation in the analysis, while others are upweighted by a factor of 10. Per Table 1, of the 3645 participants, 1.7% were positive for FPG-defined diabetes, 1.8% for impaired fasting glucose and 7.3% had $HbA_{1c} > 6.5\%$.	
4. Also, some details about log-regression FPG-defined ~ HbA1c level would be appreciated (R², F value, degrees of freedom, and significance level (p), beta-coefficients).	We have added some details to the logistic regression Results: "For every percentage unit increase in HbA_{1c} score, there was an almost 3-fold increase in the odds of having FPG-defined diabetes (OR 2.80, 95% CI 2.21-3.53, R² 0.371, Wald X²(1) 75.5, P<0.001)." The logistic regression analysis produces a Wald Chi-square statistic rather than an F statistic. The odds ratio we presented is the exponentiated form of the beta coefficient.	

VERSION 2 – REVIEW

REVIEWER	Ruth Hughes
-----------------	-------------

	University of Otago, New Zealand
REVIEW RETURNED	09-Mar-2018

GENERAL COMMENTS	Page 11 – under ‘patient and public involvement’ is it necessary to have the questions? I would favour including only the relevant answers. Page 15 – lines 44-46 I would modify to “Accordingly, evidence strongly suggests that HbA1c ‘may not’ be appropriate for detecting impaired fasting glucose ‘in this population’.” Because as the sentence stands it is overstating your findings. Also there is still the issue that you have not addressed – there is a pre-analytical decay in blood glucose levels when sampled in fluoride tubes that occurs within the first two hours following the blood draw. – you might want to include a statement to this effect in your discussion. Page 16 Line 32 ‘with’ could be replaced with ‘the’? Line 39 should this read ‘not’ well defined?
--

REVIEWER	Katherine Ogurtsova Institute for Health Services Research and Health Economics German Diabetes Center (DDZ) Leibniz Center for Diabetes Research at Heinrich Heine University Düsseldorf Germany
REVIEW RETURNED	10-Mar-2018

GENERAL COMMENTS	Dear authors, thank you very much for the update! I have no further questions.
--

VERSION 2 – AUTHOR RESPONSE

In response to your letter dated 13 March 2018, I have done the following:

- 1) I have consolidated the 'patient and public involvement' responses into a single section.
- 2) I have revised text in response to all of Reviewer #1's additional suggestions. These revisions are on pages 11, 15 and 16. I thank her for pointing out these necessary corrections.